# Potential impacts of general practitioners working in or alongside emergency departments in England: initial qualitative findings from a national mixed-methods evaluation

Arabella Scantlebury [iD] ,[1] Heather Brant,[2] Helen Anderson,[1] Heather Leggett [iD] ,[1] Chris Salisbury [iD] ,[3] Sean Cowlishaw,[4] Sarah Voss [iD] ,[2] Jonathan Richard Benger,[2,5] Joy Adamson[1]

For numbered affiliations see end of article.

**Correspondence to**
Dr Arabella Scantlebury;
arabella.scantlebury@york.ac.uk

## ABSTRACT

**Objectives** To explore the potential impacts of introducing General Practitioners into Emergency Departments (GPED) from the perspectives of service leaders, health professionals and patients. These 'expectations of impact' can be used to generate hypotheses that will inform future implementations and evaluations of GPED.

**Design** Qualitative study consisting of 228 semistructured interviews.

**Setting** 10 acute National Health Service (NHS) hospitals and the wider healthcare system in England. Interviews were undertaken face to face or via telephone. Data were analysed thematically.

**Participants** 124 health professionals and 94 patients and carers. 10 service leaders representing a range of national organisations and government departments across England (eg, NHS England and Department of Health) were also interviewed.

**Results** A range of GPED models are being implemented across the NHS due to different interpretations of national policy and variation in local context. This has resulted in stakeholders and organisations interpreting the aims of GPED differently and anticipating a range of potential impacts. Participants expected GPED to affect the following areas: ED performance indicators; patient outcome and experience; service access; staffing and workforce experience; and resources. Across these 'domains of influence', arguments for positive, negative and no effect of GPED were proposed.

**Conclusions** Evaluating whether GPED has been successful will be challenging. However, despite uncertainty surrounding the direction of effect, there was agreement across all stakeholder groups on the areas that GPED would influence. As a result, we propose eight domains of influence that will inform our subsequent mixed-methods evaluation of GPED.

**Trial registration number** ISRCTN51780222.

## Strengths and limitations of this study

► A unique primary study of 10 National Health Service case sites explores the anticipated effects of introducing General Practitioners in Emergency Departments (GPED).
► Our analysis uses a large qualitative data set and incorporates the views of multiple stakeholders.
► Data are from England only and so may not be generalisable to other healthcare settings.
► Data represent the views of those individuals who agreed to take part and so may not be exhaustive.

attendances at emergency departments (EDs) in England stood at record levels. The year 2018–2019 saw an increase of 4.4% compared with 2017–2018 and 21% since 2009–2010.[2] High levels of ED occupancy lead to crowding,[3] and this can undermine patient safety, clinical outcomes and quality of care,[3–5] delay service delivery,[6] increase associated mortality and reduce patient and clinician satisfaction.[7]

Numerous initiatives have been introduced to address the challenge of rising demand in ED attendance globally.[8–12] Examples of UK initiatives include the introduction of telephone advice and guidance (National Health Service (NHS) 111/NHS Direct) and the provision of alternative facilities (eg, walk-in centres and urgent treatment centres) for patients to access primary care for non-urgent conditions.[1 13]

It is estimated that between 15% and 40% of patients attending the ED could be treated in general practice.[14–16] Over the past decade, EDs across the UK and Europe have started to introduce general practice (GP) services in or alongside EDs.[17] In addition

## BACKGROUND

Urgent and emergency care is experiencing increasing demand globally.[1] In 2019,

to being introduced to try and tackle a rise in demand from perceived general practice patients, it was anticipated that introducing GPs in or alongside EDs would, by providing specific general practice skills and expertise, lead to improvements in patient care and control costs by reducing admission and investigation rates.[18]

In 2015, a review of NHS Urgent and Emergency Care in England proposed that selected patients should be directed to an alternative healthcare provider who could better meet their needs, thereby reducing ED attendances.[19] In 2017, this recommendation was translated into policy in the 'Next Steps on the NHS Five Year Forward View' stating that, 'Every hospital must have comprehensive front door streaming by October 2017' (p. 15).[20] To provide financial support for the introduction of GPs working in or alongside the ED, the UK government also announced a capital fund of £100 million to which hospitals in England could apply.[21–24]

Despite the recent political and financial commitment by the UK government to introducing GPs in or alongside EDs, recent guidance from the National Institute of Health and Care Excellence stated that based on current research,[25–27] there is currently 'insufficient evidence to reach a recommendation on co-located GP units'.[28] It remains uncertain how the implicit hypotheses about the effect of GPs in an ED are articulated and understood by policymakers, service leaders, health professionals and patients. These initiatives have not been subject to rigorous, independent evaluation, and there is a lack of clarity regarding the assumptions and mechanism(s) through which the predicted performance benefits for these initiatives might be achieved.[29]

In this paper, we report findings from qualitative data, which were collected as part of a wider mixed methods study evaluating the impact of GPs working in or alongside the ED (GPED). Further details of the GPED study are outlined in box 1 and in the study protocol.[29] This paper uses qualitative data from service leaders, health professionals and patients to explore the expected impact of introducing GPs into the ED to generate hypotheses that inform how GPED will be evaluated in subsequent research and implemented into practice.

## METHODS
### Design
We completed a qualitative study consisting of interviews with service leaders, health professionals and patients from 10 case study sites (table 1). The qualitative data reported here were collected as part of the wider GPED study (box 1), which was approved by East Midlands – Leicester South Research Ethics Committee (ref:17/EM/0312), the University of Newcastle Ethics Committee (Ref: 14348/2016) and also received HRA Approval (IRAS: 230848 and 218038).

---

**Box 1 The general practitioners working in or alongside the emergency department (GPED) Study**

**Objectives:** to evaluate the impact of GPED on patient care, the primary care and acute hospital team and the wider urgent care system.

**Design:** a mixed methods study consisting of three work packages.
► Work package A: mapping, description and classification of current models of GPED in all emergency departments (EDs) in England and interviews with key policymakers to examine the hypotheses that underpin GPED.
► Work package B: quantitative analysis of national data to measure the effectiveness, costs and consequences of the GPED models identified in work package A using retrospective analysis of Hospital Episode Statistics.
► Work package C: detailed mixed methods case studies of different GPED models consisting of: non-participant observation of clinical care, semistructured interviews with staff, patients and carers, workforce surveys with ED staff and analysis of locally available routinely collected hospital data.

**Patient and Public Involvement (PPI):** a study PPI group has contributed to research design and materials and data interpretation and dissemination through a series of face-to-face workshops.

**Trial status:** in progress (ISRCTN51780222).

**Funder:** National Institute for Health Research Health Services and Delivery Programme.

---

### Sampling and recruitment
Data were collected from 10 case study sites. Sites were selected purposively to ensure maximum variation according to: GPED model; GPED duration; geographical location; and deprivation index and ED volume (ED attendances).[30] Participants were sampled opportunistically by the research team, while undertaking on-site data collection. Service leaders were contacted directly via email.

### Data collection
Telephone interviews with service leaders were conducted between December 2017 and January 2018 following informed verbal consent. During interviews participants were asked to describe: their involvement in GPED and background to the policy as well as the expected impact of GPED and any potential unintended consequences online supplemental material 1.

Case study interviews with patients and health professionals were largely conducted face to face at hospital sites during GPED study data collection. Some interviews were conducted via telephone at the request of the participant. Written informed consent was provided by all participants, and all interviews were audio-recorded. Data collection took place between October 2017 and November 2018 at 10 EDs throughout England. Interviews with health professionals, patients and carers were semistructured and followed a topic guide (online supplemental material 2–7). During interviews, health professionals were asked: their current role in ED; details of their GPED model; and expected impact. Patients and carers were asked to describe why they chose to attend the ED as well as their

**Table 1**  Data collection

| | Service leaders (national) | Case studies (10 hospital sites) |
|---|---|---|
| Total number of participants interviewed | 10 | 218 (Health professionals n=124, Patients/carers n=94) |
| Interview type | Semistructured telephone interviews | Semistructured face-to-face and telephone interviews |
| Aim | In-depth understanding from key informants | In-depth understanding from selected case sites |
| Job roles represented | Department of Health and Social Care, NHS England, NHS Improvement, Royal College of Emergency Medicine | GPs working in the ED, ED doctors (juniors, registrars and consultants), nurses (streaming, triage, minor injuries and emergency nurse practitioners), ED managerial and clinical leads, and clinical directors. |

ED, emergency department; GPs, general practitioners; NHS, National Health Service.

experiences. Patients were also asked about their views on introducing GPED and its potential impact.

## Analysis

AS, HA, HL and members of the wider GPED research team undertook data collection and analysis. HA is a registered nurse with experience of working in primary care. All other members of the research team involved in data collection and analysis are health services researchers.

Analysis was facilitated by use of the qualitative data management programme NVivo. After familiarisation, a coding framework was developed through a series of roundtable discussions by the research team and was continually refined and revisited during researcher meetings on an ongoing basis throughout data collection and analysis. This framework was used to produce a series of summaries and pen portraits to describe each case site,[21] which informed a final thematic analysis during which themes were refined further for the purpose of this paper.[22] All participants and case sites were allocated unique personal IDs to protect anonymity and confidentiality. Unless otherwise specified, we use the term staff to collectively refer to GP and ED staff throughout the results section.

## Patient and public involvement

Ten public contributors with experience of using ED services have been directly involved in the design, development and interpretation of the GPED study. In addition to attending external steering group meetings and supporting the development of our original application for research funding and key study materials (eg, information sheets), our 10 public contributors have participated in regular workshops throughout the GPED study. During these workshops, public contributors were given copies of anonymised interview transcripts along with pen portraits from two of our study sites. Public contributors initially discussed how they interpreted the data, before being asked to consider whether their own interpretations resonated with the research team's framework. Additional workshops are also being held to discuss the wider GPED study's findings where both quantitative and qualitative data will be presented and discussed with the group.

## RESULTS

Service leaders and site staff perceived the national implementation of GPED as a response to increasing pressure on EDs, with a lack of supporting research evidence. Many viewed GPED as a top-down, generalised strategy that had been imposed on them without consideration of local context. Ultimately, variations in local context, ED demand and existing GP services in or alongside the ED meant it was not considered possible to implement the same system everywhere. This resulted in a 'proliferation of different models', which in turn implied that the impact of GPED on ED performance would vary substantially.

Our qualitative data highlight the challenges associated with a top-down national policy that is implemented in different ways according to local context. We hope to demonstrate the complexity and uncertainty this brings when trying to predict and then evaluate how the policy may impact patients, EDs and the wider urgent care system. Our results are therefore presented as a series of areas that stakeholders believed would be affected by the introduction of GPED and the direction of the anticipated effect.

## Performance indicators

The premise that ED staff and GPs have inherently different approaches to risk was central to the concept of GPED. GPs were perceived to frame health and illness in a different way to ED staff, with the 'wait and see' culture of primary care leading many to view GPs as more 'risk tolerant' and more appropriately qualified to care for lower acuity patients than their 'risk averse' ED colleagues. This in turn was thought to be beneficial for GPED by making GPs less likely to order unnecessary investigations, or admit or refer lower acuity patients unnecessarily, thereby reducing the time spent in the ED and enhancing patient flow. Despite this general articulation of potential performance benefits, there was significant uncertainty about the impact of GPED within the local systems included in our case studies. One of the main areas of disagreement among site staff and service leaders was whether GPs were more tolerant of risk and if so whether this would have adverse consequences for patient safety. This resulted in variation in GPED models across sites. Individual views largely varied according to the degree of integration and the specific

role of GPs within the system, making it difficult to identify generalised predictions relating to the potential impact of GPED.

## Use of investigations

Many participants were accepting of models that asked GPs to work in a hybrid ED–GP role and encouraged GPs to 'go native', becoming highly integrated within ED teams. Some models were based on the premise that GP access to investigations was crucial to GPED effectiveness, with concerns that the potential scope of GPED would be limited by GPs not being able to undertake investigations and refer to specialties. In contrast, other GPED models limited GPs to working as they would in the community, and service leaders felt strongly that for the model to run effectively, GPs and the ED should work separately. There was an idea that GPs 'going native' would encourage them to behave in a similar way to ED doctors, thereby negating any assumed benefits from GPs' different attitudes to risk, investigation and referral. Therefore, prior expectations relating to unnecessary testing were mostly factored into the GPED model at the outset.

## Hospital admissions and the 4-hour target

Reducing hospital admissions and improving performance against the '4-hour standard' (that 95% of ED patients should be discharged, admitted or transferred within 4 hours of arrival) were often quoted as among the potential benefits of GPED. However, this was not universally accepted. For example, some felt that admissions would not be affected, because the population being targeted are not those that would normally be admitted from the ED. Equally, targeting primary care patients was welcomed by ED managers, as although GP patients can be dealt with quickly in theory, in many localities, these patients are present in high volumes and were perceived to be at risk of breaching the 4-hour standard. However, some feared there might be an unintended worsening effect—diverting people with minor conditions that are theoretically quick to resolve increases the acuity of the remaining ED patient workload. If the ED is left with only high-acuity patients, there is a possibility that both the time spent in the ED and the proportion of patients who are admitted will increase, worsening the reported '4-hour' performance.

**Table 2** Arguments proposed for the potential impact of GPED on ED performance

**ED performance and performance indicators**

| Potential impact | Positive | Negative | No difference | Exemplar quote |
|---|---|---|---|---|
| Use of investigations/testing | Risk tolerant nature of GPs makes them suitable for working alongside the ED – less likely to order investigations unnecessarily. | GPs lack skills to work in ED. By 'going native' and having access to investigations/testing, GPs may lose their unique skills and work similarly to ED doctors. | Whether GPs were given access to investigations varied depending on the GPED model in place and so any impacts associated with this would be negligible. | 'It was suggested that those problems could be better dealt with by primary care clinicians who had the appropriate skills for the job and would be perhaps confident about seeing and treating and discharging without over-investigation'. (Rowan, staff interview, 07) |
| Admissions | Avoid unnecessary admissions of lower acuity patients and improve patient flow. | If the ED is left with only high-acuity patients, the proportion of ED attendances who are admitted will increase. | Admissions not affected as the population targeted is not those that would be admitted from ED. | 'But I can't pretend that I think it will make a massive difference on admissions, because the people who are waiting for admission are very largely a different group of people you see'. (Service leader interview, 02) |
| Waiting time/4-hour Key Performance Indicator | Streaming primary care patients to GP (the most appropriate clinician) reduces the risk of breaching the 4-hour target as lower acuity patients are high in volume and occupy a lot of clinician time. | Diverting patients with minor conditions who are theoretically quick to resolve will increase the acuity of ED work and make improvements in the '4-hour target' less likely. Higher acuity patients are considered more complex and so take longer to manage, increasing the potential for breaching the target. | Number of minor breaches that would need to be converted is too large to see any improvement in '4-hour performance'. | 'In theory, if you've taken all the minors, all the sort of streamed patients and minor cases out, you'll have … your staff that are there will be able to devote more time dealing with the majors. And similarly they were hoping that you'd be reducing the volume of patients coming through there but you would hopefully be able to increase the rate the patients were seen. So you would reduce the number of breach patients coming through the main ED department'. (Service leader interview, 07) |

ED, emergency department; GP, general practice; GPED, GPs working in or alongside the ED.

When stakeholders discussed possible effects of GPED on performance indicators, it was not always clear, and was not model dependent, whether GPED streamed patients were to be included or excluded from the ED figures, and assumptions regarding this influenced participants' views. Generally, performance indicators were considered blunt tools with which to evaluate impact, reflecting potential measurement issues and artefacts rather than good clinical practice. It was also anticipated that the 'visibility' and impact of GPED would be obscured by a year-on-year increase in patient attendances and hospital admissions (table 2).

### Patient outcome and experience

A process of front door 'streaming' of patients on arrival at the ED was intended to facilitate the identification of low-acuity patients and match them with the availability and skills of the treating clinician (eg, a general practitioner). This differs from 'triage' which, although often used interchangeably with streaming, refers to the identification of high-acuity patients to ensure that more urgent cases are identified and treated in a timely way. By introducing front door streaming,[31] EDs were expected to see improvements in patient outcomes (some of which are reflected in the performance standards) and experience (table 3). Streaming lower acuity patients to a GP was anticipated to improve patient care by enabling ED staff to focus on higher acuity patients and ensure that GP acuity patients are treated in GPED rather than being 'sent round the houses'. Patients were aware of the significant resourcing and financial pressures placed on the NHS and so saw value in placing GPs in the ED.

**Table 3** Arguments proposed for the potential impact of GPED on patient outcome/experience

**Patient outcome and experience**

| Potential impact | Positive | Negative | No difference | Exemplar quote |
|---|---|---|---|---|
| Streaming patients to the appropriate clinician | Improved flow of patients through the system. | Backlog created by patients having to disclose information on multiple occasions before seeing GP. | Annual growth of ED workload may mask impact of GPED on performance. | 'Intended impact was to divert as many patients who were able to be streamed to a primary care service, away from the A&E and ED departments, reducing then, surge of patients through and ensuring that patients could be seen quickly and effectively both in A&E and ED, but also in the located primary care services'. (Service leader interview, 10) |
| Patient experience | Improved patient experience by streaming patients to a GP since this avoids them being 'sent around the houses' and/or waiting in lengthy ED queues, enabling quicker assessment and discharge. | GPED patients may prevent those with higher acuity being seen in a timely manner – GPED may increase the number of patients attending ED. | | 'I'd like to think if it was working out as we'd originally envisaged that trusts would be able to flow people through the main ED departments much quicker. So we would see reduced breaches. So the four hour performance would improve but similarly patient experience would significantly improve because you would hopefully be reducing the number of delays to patients getting treated. So hopefully it would just be freeing up the ED department, by taking the streamed patients out. So that's what I was hoping we would see'. (Service leader interview 07) |
| Value of GP | Patients saw value in GPED due to resourcing and financial pressures on NHS. | GPs lack appropriate skills and experience to work in ED. | | 'What's nice is it takes the pressure off the, er, general A&E and actually emergencies can get deal with emergencies and not get clogged up'. (Teak, patient interview, 021) |

ED, emergency department; GP, general practice; GPED, GPs working in or alongside the ED; NHS, National Health Service.

There were concerns, however, from service leaders and ED staff, that patient flow could be negatively affected by GPED with a backlog created by patients being required to disclose clinical information on multiple occasions before seeing a GP or that GPED patients would prevent those with higher acuity needs being seen in a timely manner due to beliefs that GPED may increase the number of patients attending ED and associated crowding (see further).

There was strong and divided opinion between staff groups and even service leaders as to what is considered a 'GPED appropriate' patient. These opinions were often underpinned by cultural differences between GPs and ED staff and staff perceptions regarding professional competencies, boundaries and skillsets. ED staff in particular made certain assumptions about the skill set of GPs, which influenced these views. In some cases, GPs were perceived to lack the appropriate skills and experience to work in the ED, which in turn was felt to limit the potential effectiveness of GPED. Models that required GPs to 'go native' were

thought to ask GPs to work beyond their clinical competency, with some staff claiming that GPs are not up to date with ED knowledge and lacking in key clinical skills such as x-ray interpretation and suturing. There were also concerns that GPs may not recognise higher acuity patients, with associated risks to patient safety.

### Service access

There was divided opinion as to how GPED may affect ED attendance (table 4). Despite one of the aims of GPED being to create a more efficient service, both staff and patients were concerned that GPED may become a product of its own success by encouraging people to attend ED with primary care problems repeatedly and that GPED would become a replacement GP service. It was felt that despite any 'educational' component, whereby patients are encouraged to use their own GP when attending GPED, the fact that GPED guaranteed same-day access to a GP was in conflict with this message

**Table 4** Arguments proposed for the potential impact of GPED on ED attendance

**Service access**

| Potential impact | Increase | Decrease | No difference | Exemplar quote |
|---|---|---|---|---|
| GPED as a replacement primary care service | GPED becomes a replacement GP service. | | Streaming patients to most appropriate professional. Average person uses ED less than once a year so unlikely to become the main source of general practice. | 'I guess my personal view is I think they're probably putting GPs on hospitals because they've realised people are fed up of waiting to get an appointment at the GPs and they're going to hospitals, so they're not really fixing the problem there'. (Redwood, patient interview, 02) |
| Increase 'inappropriate' attendance | Same-day access to a GP may encourage 'inappropriate' attendance. | Many patients present with high acuity needs, so not the same as a walk-in centre in terms of supply. | | 'But I think, I think what it, what it does do is that, it further reinforces the concept if you've got an urgent and emergency care problem you go to ED, because not only is the ED and x-rays and prescriptions there and all the rest of it there, but now you've got primary care there as well...I kind of think it acts as a supply site driver'. (Service leader interview, 005) |
| Increase demand on ED | Peaks in attendance when general practice surgeries are closed. | | Patients unaware of GPED service. | 'It hasn't been well publicised...patients, I don't think most patients will be aware of it. I think that given they get treated in an emergency department they will probably not recognise that there is, that there's a GP service...' (Service leader interview, 01) |

ED, emergency department; GP, general practice; GPED, GPs working in or alongside the ED.

and could encourage 'inappropriate' attendance with routine rather than urgent care needs. Concerns that GPED could create additional demand on the ED were supported by anecdotal reports from established GPED models highlighting that the volume of patients had increased since introduction. This rise was attributed to the service generating new demand from primary care patients. Others highlighted the potential influence of general practice opening times; because primary care patients tend to present out of hours, GPED could cause peaks in ED attendance when general practice surgeries are closed.

Yet this view was not universal; service leaders provided various reasons why the policy was unlikely to cause an increase in ED attendance. For example, service leaders argued that given the average person attends the ED less than once a year, it is unlikely that they would start using ED as their main access to general practice. Additionally, as many ED patients present with higher acuity, GPED was not expected to be a supply driver in the same way as a walk-in centre. To this end, GPED was not viewed as being about access to GPs but about streaming patients to the most clinically appropriate professional. A lack of advertising or promotion of the availability of GPED services, the fact that most cases would still be treated in the ED and a lack of patient awareness of GPED meant that GPED eas expectedto have an egligible impact on demand.

### Staffing and workforce experience

Staffing issues dominated discussions about the potential impact of GPED and were seen to pose a major threat to its success (table 5). Services leaders and site staff expressed concern that GPED could draw GPs away from primary care and cause competition for GP staff. Consequently, GPED was perceived to have the potential to worsen general practice staffing issues, which in turn could increase waits for a GP appointment and further encourage people to attend ED.

GPED was considered an attractive prospect for those GPs seeking portfolio careers and wishing to expand their practice, knowledge and skills. Traditional general practice was seen as a more stressful and less attractive workplace than newer service models. This was due to several pressures including increasing volume and complexity of workload and depleted community and social care provision. There was some debate as to how the flexible hours associated with GPED would impact on job satisfaction. For example, some anticipated that this flexibility would make it easier to fill rotas, while others felt that shift working goes against one of the main reasons why people choose to be a GP.

Many staff perceived GPED to have training and educational benefits for junior doctors who would, in some models, become more confident about discharging patients and build up their primary care knowledge (table 6). Conversely, diverting patients with minor conditions to GPED was seen to have

**Table 5** Arguments proposed for the potential impact of GPED on staffing and experience

**Staffing and workforce experience**

| Potential impact | Positive | Negative | Exemplar quote(s) |
|---|---|---|---|
| GPs want to work 'beyond the walls of the surgery' | GPED is an attractive place to work for those wanting portfolio careers. | Working 'beyond the walls of the surgery' is not appealing to all and may cause competition for GP staff between primary and secondary care. | 'A concern [is] that it would, it would spread the primary care resource more thinly, so it would be less able to respond to, you know, would be less able to respond to sagittal primary care demand…' (Service leader interview, 05) |
| Flexible working hours | Flexible working hours may make it easier to fill rotas. | Working out of hours is a deterrent for those who chose to work in general practice. | 'Just because I'm a locum I can avoid doing nights, and chose not to do nights'. (Chestnut, staff interview, 22) |
| Locum working | Working on a locum or ad hoc basis can be attractive to some and may mitigate against GP staffing issues. | Difficult to ensure the quality of locum staff and inconsistent workforce supply negatively affects collaborative working between ED and GPs. | 'The barriers, yes. Often, the GPs are not there all the time, it's not the same person. They're often locum. So, the GP will, sort of, arrive, go straight into their room and then stay in the room unless you call them out for huddle … whereas A&E nurses and all of our doctors are all quite social, we're a team, we're really visible to each other. I think just the mentality of a GP is you sit in your room all day, don't you, on your own?' (Nutmeg, staff interview, 15) |

ED, emergency department; GP, GPs working in or alongside the ED; GP, general practice.

**Table 6** Arguments proposed for the potential impact embedding GPs in ED teams

| | Integrating GPs as part of the ED team | | |
|---|---|---|---|
| Potential impact | Positive | Negative | Exemplar quote |
| Training and clinical skills | Benefits for improving team working and skill mix. Training and educational benefits for junior doctors and GPs. | GPs may lack appropriate skills/experience to work in ED | 'Yes, knowledge and experience. GPs could teach about headaches to the primary care nurse and us, if we wanted to help out a little bit, to bring on new nurses who are coming through and learn. Then you could develop majors practitioners, bring them through. Do teaching and education, bring minors and- it would be a perfect bed of opportunity'. (Rowan, staff interview, 20) |
| Deskilling of GP and nursing workforce | Nurses prefer to work in GPED | Integrating GPs may cause deskilling. Negative views on streaming and the potential for GPED to deskill the nursing workforce by diverting minor illnesses to GPED. | 'There's a risk that the GPs who are then working on a consistent basis within an emergency department or as part of… that they can go native within that setting and actually take on more of the, qualities that you might expect to see, in other emergency department staff and actually lose the characteristics that you might expect to see of a GP'. (Service leader interview, 10) |

ED, emergency department; GP, general practice; GPED, GPs working in or alongside the ED.

benefits for ED juniors and trainees by exposing them to more acutely ill patients.

However, there was a perceived lack of suitably qualified GPs with the necessary skills and experience to work effectively in GPED. Site staff placed importance on making GPED an attractive place to work and ensuring that GPs feel valued, supported and appropriately remunerated for effective implementation. Emphasis was also placed on ensuring GPs feel protected and supported to work within their scope of practice. As a result, some felt that GPs needed to be upskilled or would require extra training. To compensate for this, some respondents emphasised the importance of recruiting experienced GPs, who had previously worked in the ED, or employing GPs that were trained at their hospital site as juniors.

There was also concern that experienced nursing staff may prefer to work in GPED due to 'better' working hours and it being perceived as an easier job. This has implications on ED staffing and on streaming, which many felt should be undertaken by an experienced nurse. However, some nurses perceived streaming to be a waste of their clinical skills and believed that it took them away from their central role and left ED short-staffed. ED nurse practitioners were also concerned that although they continued to see patients with minor injuries, minor illnesses would be streamed to GPED, which could result in deskilling of the ED nursing workforce.

### Resources

Staff and patients predicted that GPED would incur higher costs due to the cost of GP employment and placed importance on ensuring staffing and resources are carefully matched (table 7). Staff considered GPs a costly resource and felt that GPs needed to demonstrate their effectiveness. Furthermore, the employment of locums and agency staff to fill these positions was expected to lead to greater costs. There were some concerns that the funding could be better spent improving general practice provision, which may lead to the same outcome. Incidental costs such as paying for training and the set-up and management of new IT systems was considered an added cost and time burden that staff felt had not always been taken into consideration.

Positively, GPED was seen by some as a cost-effective initiative through its presumed effect of reducing hospital admissions and unnecessary patient investigations. If patients were seen by a GP, this would release ED staff to treat more unwell patients with a potential cost saving arising from the more effective use of staff resources (i.e. patients being seen by the most appropriate staff member).

**Table 7** Arguments proposed for the potential impact of GPED on resources

| | Resources | | |
| | | | |
| Potential impact | Positive | Negative | Exemplar quote |
| --- | --- | --- | --- |
| Costs | Reduction in hospital admissions and patient investigations. Streaming patients to the appropriate clinician may result in cost-savings through more effective use of staff resources. | GPs are a costly resource. Reliance on locums and agency staff. | 'Costs had a massive factor in it. Staffing, we kind of have to work around the cost. So sometimes it's, painfully, not for how many you should have to be able to run the department, it's how many can we afford to have to run the department safely'. (Chestnut, staff interview' 023) |
| Infrastructure | | Training and IT set-up and management. | 'The training was, I have to say, on the computer system, not great. I tried to get some IT training on the system. The IT department said there wasn't any training available, but they'd let me know when there was'. (Redwood, staff interview 007) |

## DISCUSSION
### Main findings

Since the 2017 implementation of 'comprehensive front door streaming', supported by capital funding,[14–18] a variety of different GPED models have been introduced throughout the NHS. This is in part a response to varying local needs and contexts, and also different interpretations of what GPED means on a practical level. This has resulted in disagreement at an individual, stakeholder and organisational level about the purpose and anticipated benefits and disbenefits of GPED and a lack of clarity about the impact of introducing GPED on these effects. Indeed, for each domain of influence, we present there were, in most cases, arguments for positive, negative and no effects of GPED (tables 2–6).

Despite disagreeing about the 'direction of effect', stakeholders agreed about which areas of the healthcare system and patient care were most likely to be impacted by GPED. This has enabled us to generate 'domains of influence', which will form the basis of our subsequent mixed-methods evaluation of the impact of GPED on patient care, the general practice and acute hospital team and the wider urgent care system during the wider GPED study (box 2).

While the domains of influence provide the foundation for our wider mixed-methods evaluation of GPED, a lack of agreement surrounding the policy's aims, coupled with uncertainty as to how the anticipated impacts will be achieved, poses a significant challenge when evaluating whether GPED can be considered a successful national policy.

It is also unclear whether the success of GPED should be determined by its effect on EDs or the wider healthcare system. This warrants careful consideration since some domains, such as ED costs or performance, may be improved at the expense of the wider NHS. Additionally, many of the differences in opinion surrounding the potential impact of GPED are underpinned by confusion as to whether patients attending the GPED are considered part of, or separate from, the denominator used for measuring ED performance. This has implications for understanding the effect of GPED on key performance indicators, particularly the '4-hour target'.

### Comparison with existing literature

In 2010, Carson *et al*[18] explored rationales for the introduction of GPED through an online survey. They report that 'The main reason was to meet the needs of patients or improve quality of care. This was followed by achieving the four-hour target and reducing cost'.

Similar assumptions have persisted and were seen to be drivers of the policy initiative to roll out GPED in all EDs across England. Benefits of GPED, particularly to address the increasing demand in emergency care, were perpetuated through rhetoric presented in the national press,[32] clinical press releases,[33] medical journals[23 34] and within the policy documents produced at the time.[35 36]

Early studies appeared to underpin some of these assumptions. Evaluations of early adopters in the UK and Europe suggested that GPs in the ED could 'result in reduced rates of investigations, prescriptions, and referrals',[9 37] increase patient satisfaction[8] and offer patients a greater range of healthcare provision.[38] However, these studies have generally been of poor quality.

---

**Box 2    GPED domains of influence**

► Performance against the 4-hour target/waiting time.
► Use of investigations.
► Hospital admission.
► Patient outcome/experience.
► Service access.
► Staffing.
► Workforce.
► Resource use/cost.

---

More recently, these assumed benefits have been challenged. A realist review concluded that despite a reduction in process time for non-urgent patients, this does not necessarily increase capacity to care for the sickest patients.[31] The main cause of ED crowding is a lack of beds and congestion in the flow of sicker patients rather than absolute attendance numbers.[39] In addition, GPED may encourage patients to present to the ED with a primary care problem, with consequent increases in ED attendance.[26 40]

To date, reviews that examine GPED in more detail have concluded that there is insufficient evidence to support national policy or local system change.[25 26 41] Two Cochrane reviews (2012 and 2018) concluded that there was 'insufficient evidence upon which to draw conclusions for practice or policy regarding the effectiveness and safety of care provided to non-urgent patients by GPs vs EPs in the ED to mitigate problems of overcrowding, wait-times and patient flow' (p. 2).[27 42]

## Strengths and limitations

The 'domains of influence' that we have identified in this paper were generated from a large evaluation that used 'big qualitative data' (228 interviews) and the views of multiple stakeholders. This provided a rich and nuanced understanding of the complexity surrounding a current national policy—GPED. Our data apply to England only, and so may not be generalisable to other healthcare settings. In addition, we could only interview those who agreed to take part, and while we did not 'strive for saturation', the range of views may not be exhaustive. However, our maximum variation approach did achieve data that span a very wide range of individuals.[30] The detail we have obtained has enabled us to propose the domains of influence that will be used to inform our wider GPED study, the aim of which is to evaluate the impact of GPED on each of the domains of influence in detail. It could be argued that the data we present here represents the inherent uncertainty and resistance to change that most healthcare policy encounters prior to or during early implementation and so is representative of typical 'teething problems.' However, while it is assumed that such issues will improve over time, recent research suggests that issues that are identified early in the implementation process often persist long after establishment.[43] It is our hope that by identifying 'domains of influence', rather than a set of hypotheses, we have mitigated against this and have identified many of the key areas that the GPED policy is likely to affect, while providing a framework to guide our forthcoming mixed methods evaluation.

## CONCLUSION

In 2017, a significant financial commitment to support hospitals introduce GPs in ED was made in a direct attempt to address growing concerns surrounding the pressures on EDs. However, the reality of introducing GPs in ED is complex. Throughout the NHS, the policy is being interpreted differently, which has created a range of GPED models to be implemented into ever-changing and variable local contexts. This variation both in terms of how the policy is being interpreted and introduced, different 'baseline levels' of GPED and the lack of agreement from stakeholders surrounding the potential benefits and dis-benefits of the policy, mean that the impact of GPED is difficult to predict. However, our findings suggest that GPED will affect eight key areas. These 'domains of influence' will be used as the foundation for our subsequent mixed-methods evaluation.

**Author affiliations**
[1]York Trials Unit, Department of Health Sciences, University of York, York, UK
[2]Faculty of Health and Life Sciences, University of the West of England, Bristol, UK
[3]School of Social and community medicine, University of Bristol, Bristol, UK
[4]Department of Psychiatry, The University of Melbourne, Melbourne, Victoria, Australia
[5]Academic Department of Emergency Care, NHS Bristol North Somerset and South Gloucestershire Clinical Commissioning Group, Bristol, UK

**Acknowledgements** The authors would like to thank the participants for their involvement in the study, the GPs working in or alongside the ED (GPED) public contributors and wider GPED research team.

**Contributors** AS drafted the manuscript, undertook data collection and analysis. HA and HL undertook data collection and analysis and critically appraised the manuscript. HB, SC, CS and SV critically appraised the manuscript. JA and JRB helped to draft the manuscript; JA also undertook analysis. JA, JRB, SC, CS and SV designed the study. All authors have reviewed and approved the final manuscript.

**Funding** This work was supported by the National Institute for Health Research (NIHR) Health Services & Delivery Research (HS&DR) Programme, project number 15/145/06.

**Disclaimer** The views and opinions expressed therein are those of the authors and do not necessarily reflect those of the HS&DR Research Programme, NIHR, National Health Service (NHS) or the Department of Health.

**Competing interests** JB is seconded part-time to the post of interim Chief Medical Officer at NHS Digital. All other authors declare no conflict of interest.

**Patient consent for publication** Not required.

**Ethics approval** Approval for the study has been obtained from the Health Research Authority (HRA) (IRAS: 230 848 and 218038). The protocol was reviewed and received a favourable opinion from the NHS East Midlands – Leicester South Research Ethics Committee REC: 17/EM/0312 and the University of Newcastle Ethics Committee (Ref: 14348/2016).

**Provenance and peer review** Not commissioned; externally peer reviewed.

**Data availability statement** All data relevant to the study are included in the article or uploaded as supplementary information. The dataset that we have acquired will not be available as our ethical approval does not permit the sharing of the entire dataset.

**ORCID iDs**
Arabella Scantlebury http://orcid.org/0000-0003-3518-2740
Heather Leggett http://orcid.org/0000-0001-8708-9842
Chris Salisbury http://orcid.org/0000-0002-4378-3960
Sarah Voss http://orcid.org/0000-0001-5044-5145

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
