## [Reviewer comments · BMJ Open]

ARTICLE DETAILS

TITLE (PROVISIONAL)	The Potential Impacts of General Practitioners Working in or Alongside Emergency Departments in England: initial qualitative findings from a national mixed-methods evaluation
AUTHORS	Scantlebury, Arabella; Brant, Heather; Anderson, Helen; Leggett, Heather; Salisbury, Chris; Cowlshaw, Sean; Voss, Sarah; Bengner, Jonathan; Adamson, Joy

VERSION 1 – REVIEW

REVIEWER	Kelly, Shona Sheffield Hallam University, Faculty of Health and Wellbeing
REVIEW RETURNED	19-Nov-2020

GENERAL COMMENTS	This is a fascinating study and raises all sorts of issues that won't fit in this BMJ Open paper. I have very few comments to make for minor revisions. 1. I think you don't make enough of the obvious cultural divide between primary and acute care.2. There are a few typos and page breaks that fall in the wrong place. Additional comments that you should consider for the Background or Discussion 3. Is the UK, the only country in the world with GPs in ED? I remember reviewing papers from the Netherlands on this and I see you quote Boeke. And also a dim memory of primary care being an outpatient department in Japan or Korea. Since 1982, Canada has offered family doctors emergency department experience and now some have emergency medicine certification (https://www.ncbi.nlm.nih.gov/pmc/articles/PMC128398/) I think there is some confusion in the UK about Out-of-hours service provision (or the lack of it) and EDs. I realise the word count is tight but some background for an international audience would increase the number of citations.4. This is an enormous number of interviews and I think you should explain earlier that is because there is no framework under which this national initiative was rolled out. A typical English approach I'm afraid. Do you feel you reached data saturation?5. Is there a difference between "streaming" and triage?6. The comments about exactly who is a GPED appropriate patient highlights the divide that exists within the NHS between Primary and Acute Care. The NHS states that GPs "... focus on the health of the whole person combining physical, psychological and social aspects of care" but this doesn't appear to have been understood by your ED interviewees. Nor do they recognise the magnitude of the non-traumatic care that they are providing in ED. The GPs whole person approach places GPs in a much better position to manage the non-trauma ED patient. I think there is a degree of
---

	snobbery operating here. Note the Canadian approach I highlighted above where there is also a national healthcare system and a shortage of GPs 7. Did your findings shed any light on the local Primary Care – Acute Trust relationships? Were there any?
--	--

REVIEWER	Morley , Claire University of Tasmania
REVIEW RETURNED	22-Dec-2020

GENERAL COMMENTS	Thank you for the opportunity to review this very well written manuscript. A key strength of the manuscript lies in the way the authors have clearly explained in the introduction the difficulties with evaluating implementation of this policy initiative; namely the apparent lack of direction as to the intended outcomes. Notwithstanding, the authors have done a thorough job of engaging all of the key stakeholders to try to gain an understanding of the actual and perceived outcomes. I think the final evaluation based on the domains of influence identified in this study, coupled with the planned quantitative analysis (Work Package B) will be necessary and valuable work to inform the future of this initiative. I wish you well. My only suggestion for improvement would be in the Results section. Whilst Table two adds additional information to the proceeding text, I found Tables 3-6 (noting there are 2 table 3's) rather repetitive, as outside of the direct quotes they did not add new information not covered in the proceeding text. I would consider removing these tables. Best of luck with this important research.
---

VERSION 1 – AUTHOR RESPONSE

Reviewer 1	Response
This is a fascinating study and raises all sorts of issues that won't fit in this BMJ Open paper. I have very few comments to make for minor revisions.	Thank you we are really pleased to hear that you enjoyed reading the paper.
1. I think you don't make enough of the obvious cultural divide between primary and acute care.	We agree that this is an important issue and one that is worthy of comment. We have added some text to the results section (patient outcome and experience; lined 277-280) to highlight that cultural divides between primary and acute care may have been influencing opinions on the potential impacts of GPED. We have done this in a way that does not retrospectively place emphasis on certain issues, as the data which we report

	here is based on what was of key importance to our participants.
. There are a few typos and page breaks that fall in the wrong place. Additional comments that you should consider for the Background or Discussion	We apologise for this and have tried to rectify these as much as possible ahead of the 'copy editing' stage, should the manuscript be accepted.
3. Is the UK, the only country in the world with GPs in ED? I remember reviewing papers from the Netherlands on this and I see you quote Boeke. And also a dim memory of primary care being an outpatient department in Japan or Korea. Since 1982, Canada has offered family doctors emergency department experience and now some have emergency medicine certification (https://www.ncbi.nlm.nih.gov/pmc/articles/PMC128398/) I think there is some confusion in the UK about Out-of-hours service provision (or the lack of it) and EDs. I realise the word count is tight but some background for an international audience would increase the number of citations.	Thank you for this suggestion and for your understanding surrounding word count. We have added some additional information to our background section, which we feel gives a clearer picture of the key policies/literature on the introduction of GPs in EDs and associated initiatives within the UK and internationally.
4. This is an enormous number of interviews and I think you should explain earlier that is because there is no framework under which this national initiative was rolled out. A typical English approach I'm afraid. Do you feel you reached data saturation?	In keeping with current methodological guidance we deliberately chose not to 'capture' data saturation. Instead, our aim was to ensure that we obtained maximum variation in our sample according to: types of GPED model, GPED duration, geographical location, deprivation index, ED volume (Case sites); a range of staff groups, grades, specialties and patients presenting with different conditions, of different ages, genders. We do feel that our data is representative of a varied sample. We have added the below reference to the sampling and recruitment section (methods) and some text to clarify our position on saturation to the strengths and limitations section (discussion). Braun, V., & Clarke, V. (2019). To saturate or not to saturate? Questioning data saturation as a useful

	concept for thematic analysis and sample-size rationales. Qualitative Research in Sport, Exercise and Health, 1-16.
5. Is there a difference between “streaming” and triage?	Yes, we have added text to make this clearer in the patient outcome and experience section (line 260-263).
6. The comments about exactly who is a GPED appropriate patient highlights the divide that exists within the NHS between Primary and Acute Care. The NHS states that GPs “... focus on the health of the whole person combining physical, psychological and social aspects of care” but this doesn’t appear to have been understood by your ED interviewees. Nor do they recognise the magnitude of the non-traumatic care that they are providing in ED. The GPs whole person approach places GPs in a much better position to manage the non-trauma ED patient. I think there is a degree of snobbery operating here. Note the Canadian approach I highlighted above where there is also a national healthcare system and a shortage of GPs	We agree that this finding highlights the cultural divide between primary and secondary care and have added additional text to the results section (patient outcome and experience lines 257-262) to strengthen this point further.
7. Did your findings shed any light on the local Primary Care – Acute Trust relationships? Were there any?	Interestingly, this subject did not arise beyond what is reported surrounding concerns over worsening the primary care crisis (Results: Staffing and workforce experience).
Reviewer 2	Response
Thank you for the opportunity to review this very well written manuscript. A key strength of the manuscript lies in the way the authors have clearly explained in the introduction the difficulties with evaluating implementation of this policy initiative; namely the apparent lack of direction as to the intended outcomes. Notwithstanding, the authors have done a thorough job of engaging all of the key stakeholders to try to gain an understanding of the actual and perceived outcomes. I think the final evaluation based on the domains of influence identified in this study, coupled with the planned quantitative analysis (Work Package B) will be necessary and valuable work to inform the future of this initiative. I wish you well.	Thank you
My only suggestion for improvement would be in the Results section. Whilst Table two adds additional information to the proceeding text, I found Tables 3-6 (noting there are 2 table 3's) rather repetitive, as outside of the direct quotes they did not add new information not covered in the proceeding text. I would consider removing these tables.	Thank you for this suggestion. We have corrected the way that our tables are numbered throughout the manuscript. We do however feel strongly that the tables are important and should remain as they

	are. When drafting the manuscript, there was much discussion amongst the team and wider project group regarding the various ways that we could present our findings for this manuscript. Our rationale for the tables was mainly to illustrate the range of topics and often divided and contradictory opinions that were reported by our participants. As reviewer 1 alludes to, some of the arguments and opinions raised by participants are not always what people may expect and so we think it is essential that they are reflected here as they give important context to the main body of the text.
--	--

VERSION 2 – REVIEW

REVIEWER	Kelly, Shona Sheffield Hallam University, Faculty of Health and Wellbeing
REVIEW RETURNED	18-Feb-2021

GENERAL COMMENTS	Good work. I look forward to referencing it in my research
--

REVIEWER	Morley , Claire University of Tasmania
REVIEW RETURNED	15-Apr-2021

GENERAL COMMENTS	Thank you to the authors for revising the manuscript as per reviewers and editors comments. I think the findings will generate a lot of interest. I look forward to reading the publication.
---